# Study Protocol for a Multicenter, Randomized Controlled Trial to Improve Upper Extremity Hemiparesis in Chronic Stroke Patients by One-to-One Training (NEURO^®^) with Repetitive Transcranial Magnetic Stimulation

**DOI:** 10.3390/jcm11226835

**Published:** 2022-11-18

**Authors:** Daigo Sakamoto, Toyohiro Hamaguchi, Kai Murata, Atsushi Ishikawa, Yasuhide Nakayama, Masahiro Abo

**Affiliations:** 1Department of Rehabilitation Medicine, The Jikei University School of Medicine Hospital, Tokyo 105-8471, Japan; 2Department of Rehabilitation, Graduate School of Health Science, Saitama Prefectural University, Saitama 343-8540, Japan; 3Department of Rehabilitation Medicine, The Jikei University School of Medicine, Tokyo 105-8461, Japan

**Keywords:** occupational therapy, upper limb function, study protocol, stroke

## Abstract

During recovery from upper limb motor paralysis after stroke, it is important to (1) set the exercise difficulty level according to the motor paralysis severity, (2) provide adequate exercises, and (3) motivate the patient to achieve the goal. However, these factors have not been well-formulated. This multicenter, randomized controlled trial study aims to examine the therapeutic effects of these three factors on patients undergoing a novel intervention using repetitive transcranial magnetic stimulation and intensive one-to-one training (NEURO^®^) and to formulate a corresponding research protocol. The control group will receive conventional NEURO^®^ occupational therapy. In the intervention group, four practice plans will be selected according to the Fugl-Meyer assessment (FMA-UE) scores of the upper extremity. The goal is to predict the post-treatment outcomes based on the pre-treatment FMA-UE scores. Based on the degree of difficulty and amount of practice required, we can formulate a practice plan to promote upper limb motor recovery. This occupational therapy plan will be less influenced by the therapist’s skill, facilitating effective rehabilitation. The study findings may be utilized to promote upper limb motor paralysis recovery and provide a basis for proposing activities of daily living adapted to upper limb function.

## 1. Introduction

Approximately 80% of stroke patients develop motor paralysis [1]. Motor paralysis of the upper extremity limits patients’ activities of daily living (ADL) and reduces their quality of life [2,3]. To improve the motor function of the upper extremity in stroke patients, it is necessary to induce neuroplasticity in the patient’s brain through continuous rehabilitation [4,5,6]. Novel intervention using repetitive transcranial magnetic stimulation (rTMS) and intensive one-to-one training (NEURO^®^) is one such approach. 

The role of rTMS in NEURO^®^ is to pre-condition for rehabilitation by regulating movement-related neural activation. The effects of rTMS depend on the location, frequency, and intensity of the stimulation [7,8]. For example, in transcranial magnetic stimulation in NEURO^®^, the primary motor cortex of the patient’s intact cerebral hemisphere is irradiated at 2400 pulses per day [6]. The stimulation intensity is defined as 90% of the resting motor threshold, with the first dorsal interosseous muscle of the non-paralyzed side as the target muscle. rTMS attenuates abnormal muscle tone by decreasing the excitability of cortical and spinal anterior horn cells and modulating interhemispheric inhibition [9,10,11]. For stroke patients in the acute and convalescent (with stable symptoms 1 or 2 months after onset) phases, methods have been developed to increase neural excitability by applying high-frequency rTMS (>4 Hz) to the affected cerebral hemisphere [12,13,14]. Methods have also been developed to stimulate the bilateral cerebral hemispheres [15]. Rehabilitation in NEURO^®^ is intended to promote use-dependent plasticity of the brain by repeated motor exercises under the condition that the patient’s interhemispheric inhibition is regulated by rTMS. Occupational therapists instruct the patients on NEURO^®^ regarding the (1) functional motor exercises in the proximal and distal parts of the upper limb, (2) movement exercises including reaching and manipulating objects, (3) ADL exercises using the paralyzed side, and (4) ADL instruction and self-exercises to encourage the usage of the paralyzed side in daily life [4,5,6].

The goal of patients treated with NEURO^®^ is to reacquire ADL using the paralyzed side. In occupational therapy, compensatory movements using the non-paralyzed side, assistive devices, and assistive technology may be applied to supplement the patient’s motor function to reacquire the target activities [16,17,18]. However, in some cases, improving the motor function of the affected side may increase the possibility of patients acquiring the desired ADL. A hemiplegic patient needs at least 20 h of practice during a two-week hospitalization to recover upper limb motor function [19]. The number of stimulated upper limb movements is estimated to be 420 per session [19,20]. Rehabilitation is also more effective when patients are aware of their goals for the recovery of motor function and achievement of ADL [21,22]. Clinical occupational therapists presumably share the treatment goals with patients based on the physician’s prediction of recovery after treatment and provide selected exercises in appropriate amounts according to the characteristics of individual patients’ motor paralysis. The motor function of patients undergoing NEURO^®^ can be predicted using the Fugl-Meyer assessment of the upper extremity (FMA-UE) score [23]. NEURO^®^ reportedly improves upper limb motor function in patients regardless of the stroke type [24]. The content of rehabilitation provided by the therapist presumably affects the patient’s recovery; however, this has not been verified.

Occupational therapists intervene with the aim of helping patients acquire their target movements. They induce neuroplasticity by rTMS after considering the severity of motor paralysis. In previous studies on NEURO^®^, occupational therapy exercises were determined by the physician and therapist-in-charge with the consent of the patient, and 40 to 60 min of treatment were provided twice a day [4,5,6]. However, when compared with the well-established stimulation method of rTMS, the difficulty level, amount of practice, and goal settings in NEURO^®^ are not yet formalized and are dependent on the discretion of the physicians and therapists. Motor paralysis and related disability in daily life are highly individualized; while it is difficult for patients to have an idea of the training and recovery needed, this is not a problem for therapists with sufficient clinical experience. The issues faced include: (1) unclear criteria for selecting upper limb motor exercises based on the severity of motor paralysis, (2) unavailability of guidelines for the number of upper limb exercises to be performed by patients, and (3) unclear goal-sharing between patients and therapists regarding the recovery of upper limb motor function and the use of upper limbs in ADL. Therefore, a formalized occupational therapy in NEURO^®^ will uniformize the treatment method and stabilize the functional recovery of patients.

This study aims to evaluate the therapeutic effect of NEURO^®^ on patients by providing a certain amount of selected functional exercises based on the severity of the patients’ motor paralysis. We also define a research protocol in this study. It is hypothesized that the occupational therapy intervention used in this study will improve the patients’ upper limb motor function compared to the conventional intervention method used for patients who receive NEURO^®^. We believe that if the treatment plan for NEURO^®^ occupational therapy is established according to the characteristics of the patients’ motor paralysis, the patients will be less influenced by the skills and years of experience of the assigned occupational therapist, and the effectiveness of rehabilitation will be more stable than that of conventional occupational therapy. This stability will contribute to the improvement of the patient’s goal for ADL.

## 2. Materials and Methods

### 2.1. Aims

The purpose of this study is to evaluate the therapeutic effect of NEURO^®^ on patients by providing a certain amount of selected functional exercises based on the severity of the patients’ motor paralysis. We also aim to define a research protocol in this study.

### 2.2. Study Design

We will use a multicenter, randomized controlled trial design to conduct this study.

### 2.3. Participants and Settings

The study participants are patients with a history of stroke who are admitted to a NEURO^®^-certified facility and have undergone NEURO^®^. NEURO^®^ is performed in 14 registered NEURO^®^-accredited facilities by trained staff who have passed the examination of the Department of Rehabilitation Medicine, The Jikei University School of Medicine [25]. The inclusion criteria for patients, based on the rTMS guidelines, are as follows: (1) hemiplegia after first stroke, (2) age ≥ 20 years, (3) 6 months after stroke onset, and (4) no bilateral cerebrovascular disease [26]. Exclusion criteria are as follows: (1) patients with a diagnosis of dementia based on a Mini Mental State Examination score below the cutoff point, (2) patients with physical or psychiatric diseases requiring medical management, (3) patients with seizures within 1 year, (4) patients with intracranial clips or cardiac pacemakers, (5) patients with a history of subarachnoid hemorrhage, and (6) patients with fever (≥37 °C), upper respiratory tract inflammation, malaise, and taste or olfactory symptoms when seen by a rehabilitation physician affiliated with a NEURO^®^ -accredited facility prior to admission.

### 2.4. Participant Characteristics

Participant characteristics such as age, sex, height, weight, body mass index (BMI), and dominant hand will be investigated. Medical information regarding the type of disease (cerebral infarction, cerebral hemorrhage, etc.), paralyzed side, post-onset period, and the years of clinical practice of the treating therapists will be investigated (Table 1).

### 2.5. Sample Size

The sample size was estimated via a priori analysis using G*Power. We selected goodness-of-fit (F test) and analysis of covariance (ANCOVA) (for fixed effects, main effects, and interactions) for finding the sample fitness and size. The other data were set as α = 0.05, 1 − β (power) = 0.8, numerator df = 1, number of groups = 4, and number of covariates = 4. Based on this calculation, the minimum sample size was estimated to be 128 (64 in each group). Since the subjects were hemiplegic patients, the minimum sample size was defined as 128 subjects (64 subjects in each group).

### 2.6. Randomization

A random number table, created using Microsoft Excel (Microsoft, Redmond, WA, USA), will be used for the allocation of patients into the intervention and control groups. The random number table is created at each hospital and maintained by the allocation manager at each hospital.

### 2.7. Sequence Generation

Simple randomization will be used for allocation. Once patients who meet the eligibility criteria are enrolled, they will be assigned using a random number table generated at each hospital.

### 2.8. Allocation Concealment Mechanism

Allocation will be performed by random assignment using a table of random numbers created at each hospital. The order of patient allocation will not be concealed from the person in charge of allocation. Patients will be assigned in the order in which they are registered according to the random number table (Figure 1).

### 2.9. Blinding

This study will be double-blind. (1) Patients who meet eligibility criteria and are enrolled will be blinded to the treatment received and to the group to which they are assigned. Physicians and occupational therapists will not be blinded to the group assignment. The evaluator is not blinded. (2) The data will be analyzed by an analyst using a dataset that has been processed in such a way that the group assignment is unknown. In order to confirm the success or failure of the patients’ blinding, therapists will ask patients which treatment they thought they received after completing NEURO^®^ treatment and then confirm whether it was the treatment actually performed.

### 2.10. Interventions

#### 2.10.1. rTMS

All patients undergoing NEURO^®^ will be hospitalized for 2 weeks. The rTMS will be directed at the primary motor cortex of the patient’s healthy cerebral hemisphere at 2400 pulses a day at a low frequency of 1 Hz, as in the conventional NEURO^®^ protocol (Table 2). The rehabilitation physicians affiliated with NEURO^®^-accredited facilities who are in charge of the patient will administer rTMS. The stimulation intensity is set at 90% of the resting motor threshold for the first dorsal interosseous muscle of the non-paralyzed side. This is the lowest intensity that activates motor-evoked potentials in a muscle. A figure-8 coil with a 70 mm diameter and a Mag Pro R 30 stimulator (Mag Venture Company, Farum, Denmark) are used for stimulus irradiation. The duration of treatment with NEURO^®^ and the method of rTMS irradiation are not changed between the control and intervention groups.

#### 2.10.2. Rehabilitation

Rehabilitation, as in the previous NEURO^®^ protocols, will consist of up to six 20 min intervention sessions per day for all patients, except on Sundays and the days of admission and discharge. Rehabilitation in NEURO^®^ is delivered according to prescribed methods by therapists who practice NEURO^®^ at certified medical institutions. Occupational therapy will be provided for a minimum of three sessions per day. The assignment of occupational therapy and physical therapy sessions will be determined by the attending physician with the patient’s consent, considering the patient’s needs, goals, and physical function. Moreover, all patients will be required to perform 60 min of independent training twice a day [4,5,6]. The method of assigning occupational and physical therapy sessions will be kept consistent between the control and intervention groups. In this study, the goals of occupational therapy and the content of the exercises provided in the intervention group will be defined.

### 2.11. Intervention for the Control Group

The control group will receive occupational therapy as per conventional NEURO^®^ [4,5,6]. Functional exercises, such as proximal and distal upper extremity joint movements, muscle mobilization, movement exercises including reaching and manipulation of objects, and daily living exercises to promote the patient’s goals and use in daily life will be performed. The patient will be encouraged to use the paralyzed side of the body for self-training and to manage daily life situations.

### 2.12. Treatment for the Intervention Group

In the intervention group, (1) the difficulty level of practice will be set according to the severity of motor paralysis, (2) the amount of practice necessary for the recovery of motor paralysis will be set, and (3) the treatment goals of the patient and therapist will be set with regard to the functional practice performed in occupational therapy. Movement practice including reaching and manipulation of objects and daily living practice to promote the patient’s goals and use in daily life will be performed using the same intervention methods as in conventional occupational therapy [4,5,6].

The exercise difficulty level will be selected from four plans, according to the scores of shoulder flexion to 90° during elbow extension in Part A and finger extension in Part C of FMA-UE (Figure 2). In studies investigating the hierarchy of difficulty levels of FMA-UE, the synergic movement difficulty level was lower than the difficulty level in shoulder flexion to 90° during elbow extension [27,28,29]. The ability or inability to perform shoulder flexion to 90° during elbow extension is considered a criterion to determine whether a patient is able to perform movements other than the synergic movement (Figure 3). Shoulder flexion to 90° during elbow extension and finger extension reportedly play a role in the functional prognosis of motor paralysis [30,31,32]. In the case of ADL, reaching movements to the target body part or object are planned, followed by skillful finger movements. Therefore, the patient’s ability to perform shoulder flexion to 90° during elbow extension is considered an indicator of whether the practice of the proximal or distal part of the upper limb should be prioritized. Each plan is assigned a menu of exercises with reference to studies that have examined the hierarchy of difficulty levels in the sub-items of FMA-UE [27,28,29]. Each plan includes five exercises for one-to-one practice with the occupational therapist and self-training for the patient (Table 3). We will introduce one of the five functional exercises after a discussion between the patient and therapist and after obtaining the patient’s agreement on the exercises that are highly relevant to their daily activities and those that are requested by the patient. Different training menus may be set for one-to-one training and self-training. However, they cannot be changed during the hospitalization period or mid-session. The exercises can be divided into the following three types, depending on the amount of assistance required and load of the exercise: (1) exercises to move the paralyzed limb with resistance (active resistive exercises), (2) exercises to move the paralyzed limb without resistance or assistance (active free exercises), and (3) exercises to move the paralyzed limb with manual guidance from the therapist or with assistance from the patient’s non-paralyzed limb (active assistive exercises). The exercises are categorized based on whether the joints are moved by the therapist or external forces (passive exercise) [33,34]. In the present study, exercises including voluntary movements are assigned under four plans. Passive exercises, such as stretching to decrease muscle tension and joint exercises to increase range of motion are not included in these four plans. Passive exercises are to be performed by the therapist while conditioning the patient’s body prior to the functional exercises or when relaxing between treatments, as appropriate.

The patient’s posture during practice and the load of the exercise are determined according to the patient’s physical function, such as motor paralysis, muscle tone and balance, and the purpose of the exercise to be performed. The occupational therapist is not allowed to change the content of the set joint exercises but is allowed to change the posture and the amount of load during the exercises according to changes in the patient’s muscle tone, muscle output, and motor paralysis. Figure 4 shows the method for determining the posture, and Figure 5 shows the method for determining the practice load.

The upper-extremity movements will be performed 100 times each per day for a total of 500 times, from a menu of five exercises. These exercises should be performed on a one-to-one basis between the occupational therapist and the patient. The number of upper-extremity movements in self-training will not be included in the number of five exercises. The patient is expected to perform a total of 500 joint exercises per day, including occupational therapy for functional exercises, except on Sundays, the day of admission, and the day of discharge. For example, when a patient is admitted on Monday and discharged on the following Saturday, they will perform a minimum of 5000 joint movements for functional exercises (Table 2).

We estimate that patients will perform more than 5000 upper-extremity movements during their hospitalization because, in addition to regular training, functional exercises, object handling practices, and ADL training will be scheduled. Peurala et al. reported that during a two-week period of hospitalization in which constraint-induced movement therapy was performed, a practice period of 20–56 h was effective in improving upper limb function [20]. Han et al. simulated the number of reaching movements to increase voluntary use of the paralyzed upper limb and reported that 420 movements per session were the threshold [19]. All patients undergoing NEURO^®^ will receive two hours of therapist-led rehabilitation and two hours of patient-led training per day, except on the day of admission, the day of discharge, and Sundays, during a two-week inpatient treatment period.

The minimum number of functional exercises in the intervention group will be 500 per day. The protocol of the intervention group meets the requirements of the amount and duration of exercises reported in previous studies. This study is planned in 14 NEURO^®^-certified facilities in Japan: The Jikei University Hospital, The Jikei University Daisan Hospital, Tokyo General Hospital, Nishi-Hiroshima Rehabilitation Hospital, Kimura Hospital, Kyoto Ohara Memorial Hospital, Izumi Memorial Hospital, Hakodate Shintoshi Hospital, Aomori Shintoshi Hospital, Hattanmaru Rehabilitation Hospital, Atsuchi Rehabilitation Hospital, Ainomiyako Neurosurgery Hospital, Shinagawa Rehabilitation Hospital, and International University of Health and Welfare Ichikawa Hospital. A total of 15 professionals with at least five years of experience in NEURO^®^ rehabilitation were selected as therapists. The level of difficulty of functional exercises in the occupational therapy intervention used in this study was discussed with them and decided upon via consensus. In the past, the treatment and evaluation of patients undergoing NEURO^®^ have been performed by occupational therapists who have completed the prescribed training at NEURO^®^-accredited facilities. The treatment and evaluation procedures for the intervention group will be fully explained to the occupational therapists by at least one expert selected at each site who has discussed this study protocol with the occupational therapists. The expert at each site will supervise the occupational therapists to ensure that they fully understand and implement the study protocol for the intervention group.

The treatment goal is set according to the study by Hamaguchi et al. [23], and the predicted value of recovery of upper limb motor function by NEURO^®^ is calculated using the FMA-UE score obtained on the first day of hospitalization [23]. The physician-in-charge will explain to the patients the predicted score at discharge, the target ADL, and the plan of functional exercises to be performed to achieve the goal. They also promote the patient’s understanding of these plans. These procedures are performed by the physician and occupational therapist within two days of admission.

### 2.13. Outcome Evaluation

The outcomes will be evaluated primarily by FMA-UE. FMA-UE is a comprehensive battery of tests that examines motor function, balance, range of motion of the joints, and degree of joint pain in stroke patients [35]. The motor function items of the upper limb are scored on a 66-point scale using a 3-point ordinal scale. Joint and isolated movements are evaluated in accordance with the recovery stage of motor paralysis. The total score of FMA-UE is classified into no (<23), poor (≤23–≤31), limited (≤32–≤47), notable (≤48–≤52), and full (≤53–≤66) capacities using the severity reported by Hoonhorst et al. [36].

The secondary assessment is an Action Research Arm Test (ARAT). ARAT is an upper extremity functional assessment tool based on the upper extremity test [37,38]. It consists of four sub-items and includes tasks of grasping and carrying objects, manipulation, and reaching to one’s own body. It is scored on a 57-point scale using a 4-point ordinal scale. As a second secondary evaluation, the Wolf Motor Function Test (WMFT) will be used. The WMFT consists of six tasks involving upper-extremity movements and nine tasks of object manipulation [39]. In the WMFT, the performance time of each task is measured, and the quality of movement is scored using a 6-level ordinal scale. In addition, the Jikei Assessment Scale for Motor Impairment in Daily Living (JASMID) will be used to examine the use of the affected upper limb in daily life [40]. JASMID is a patient-reported outcome that was developed based on Motor Activity Log and adapted to the Japanese lifestyle [41,42] and has been used in a previous NEURO^®^ study [23]. JASMID consists of 20 questions related to upper limb movement and evaluates the frequency of use and quality of movement of the upper limb on the affected side on a 5-point ordinal scale. The FMA-UE and ARAT will be used to measure the therapeutic effects of mild motor paralysis, which may be underestimated due to the ceiling effect. For patients with mild symptoms, the WMFT and JASMID will be used to evaluate the treatment effect using the task performance time, frequency of use of the upper extremity on the affected side, and quality of movement [43,44]. Since sleep duration influences the promotion of neuroplasticity in the brain, we investigate patients’ sleep duration and sleep quality using a questionnaire [45]. Patients are asked to respond to the sleep quality questions using a 3-point scale (1: did not sleep well, 2: unsure, 3: slept well).

### 2.14. Statistical Analysis

A multivariate ANCOVA will be conducted to test the hypothesis that the occupational therapy intervention in this study will improve motor function in chronic stroke patients undergoing NEURO^®^ when compared with conventional occupational therapy interventions. The dependent variable will be the date or period of assessment, and the independent variables will be the change in scores of the primary assessment (FMA-UE) and the secondary assessments (ARAT and JASMID). The covariates are age, sex, BMI, and time since onset. JAMOVI version 2.2.1 (JAMOVI Project, Sydney, Australia) will be used for statistical analysis.

The primary analysis will be conducted when the intervention and control group samples reach a total of 64 cases each in all settings. The secondary analysis will be conducted when the number of patients in each group exceeds 60% of all participating centers (Figure 6).

### 2.15. Ethical Considerations and Declarations

All patients will provide written consent to participate in this study. This study was approved by the Jikei University School of Medicine Ethics Committee (approval number 32-33810423). The study has been registered in the Clinical Trials Registry of the University hospital Medical Information Network (UMIN) Center (UMIN Test ID: UMIN000047489).

Doctors and occupational therapists will wear masks, wash their hands and sterilize with alcohol when providing treatment.

### 2.16. Status and Timeline of the Study

This study was approved by the ethics committee and registered as a clinical trial. This study will begin in December 2022 and end in August 2024.

## 3. Discussion

We predict that the occupational therapy interventions in this study will improve patients’ motor function compared to conventional occupational therapy interventions. The occupational therapy intervention in this study specifies (1) the level of practice difficulty, (2) the amount of practice, and (3) the method of goal setting. (1) The difficulty levels of the exercise menus in the four plans are set according to the item-specific difficulty levels of FMA-UE. Functional exercises are selected in accordance with the severity of the patient’s motor paralysis [27,28,29]. (2) The amount of practice is specified with reference to studies that have verified the threshold of the number of joint movements required for the recovery of motor paralysis [19]. Patients will be provided with an adequate amount of practice according to the selected practice plan. (3) Goal setting is based on the FMA-UE scores at the initial evaluation to predict recovery; the treatment goals and the selected practice plan will be explained to the patients [23]. One of the five functional exercises will be introduced based on the patient’s preference. The patient’s understanding of the set goals and the performance of active practice are reportedly important for the recovery of motor paralysis [21,22]. Therefore, the occupational therapy intervention in this study can promote the use-dependent plasticity of patients’ brains more effectively than conventional occupational therapy interventions and is expected to improve patients’ motor functions.

If the occupational therapy intervention used in this study improves the patients’ upper limb motor function when compared to that of the conventional occupational therapy intervention, the level of practice difficulty and the amount of practice used in this study can act as a guideline to formulate a new practice method to improve the upper limb function of patients. Occupational therapists can provide a sufficient amount of joint movement exercises according to the severity of the patient’s upper limb motor paralysis to encourage use-dependent plasticity in the patient. Standardization of occupational therapy treatment content and equalization of treatment allow efficient and effective rehabilitation of patients. Improvement in motor function contributes to the improvement of the patient’s ADL (Figure 7). The results obtained after the implementation of this protocol will assure that the ADL and certain results of occupational therapy can be attained by patients with upper motor paralysis, in any facility.

This study has a few limitations. Since this study will be conducted at a facility in Japan, it is unclear whether similar results will be obtained in other countries. We speculate that there is a limitation to the generalization of this study’s results. In countries other than Japan, the medical systems are different, and it is difficult to implement the same treatment methods since insurance for hospitalization of patients with chronic stroke and the roles of doctors and therapists vary. There are also cultural differences in the ADL required by patients. Multinational studies are needed to clarify this point. In addition, the patient’s nutritional status may affect the recovery of motor paralysis. In this study, it will not be possible to obtain data on the nutritional status of patients prior to inpatient treatment.

## Figures and Tables

**Figure 1 jcm-11-06835-f001:**
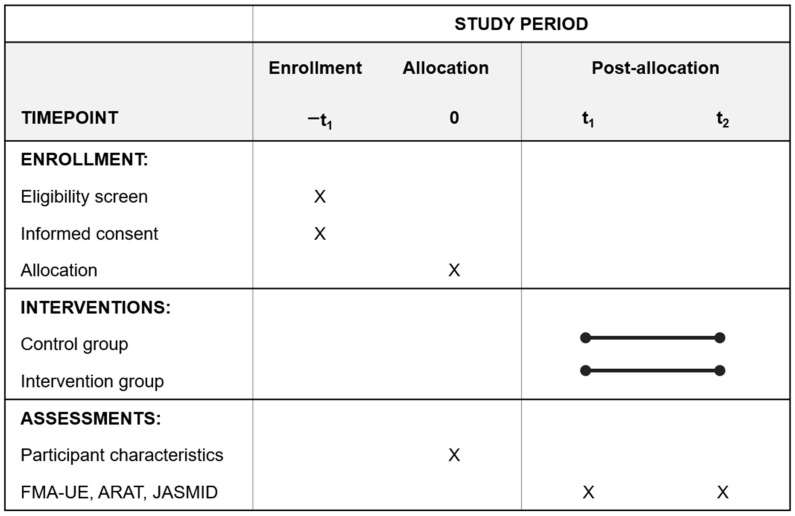
Schedule of enrollment, interventions, and assessments. FMA-UE, Fugl-Meyer assessment of the upper extremity; ARAT, Action Research Arm Test; JASMID, Jikei Assessment Scale for Motor Impairment in Daily Living. Investigators will obtain informed consent from patients who meet the eligibility criteria. Patient allocation will be performed by random assignment using a table of random numbers created at each hospital. Participant characteristics and medical information will be investigated. The control and intervention groups will receive treatment for two weeks. Patients will be evaluated on the day of admission and on the day of discharge.

**Figure 2 jcm-11-06835-f002:**
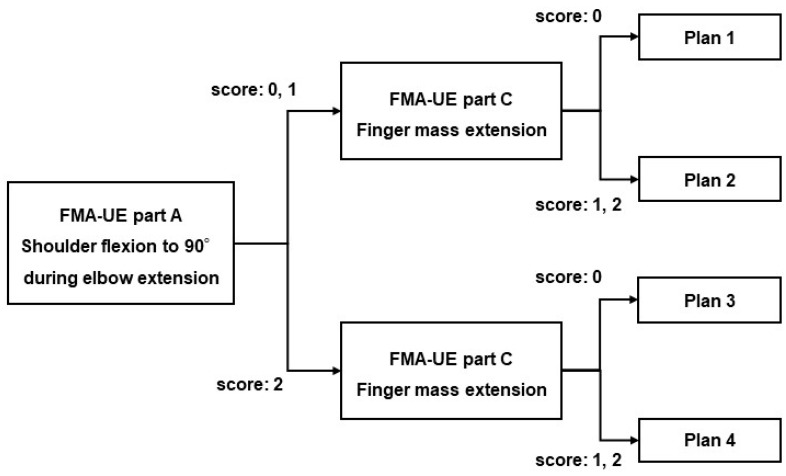
Chart of procedures for determining a functional practice plan. FMA-UE, Fugl-Meyer assessment of the upper extremity. The FMA-UE Part A (shoulder flexion to 90° during elbow extension) and Part C (finger mass extension) scores, evaluated on the day of admission, are used to determine the plan of functional exercises. The functional exercise plans are classified from 1 to 4 according to the scores from the two tests.

**Figure 3 jcm-11-06835-f003:**
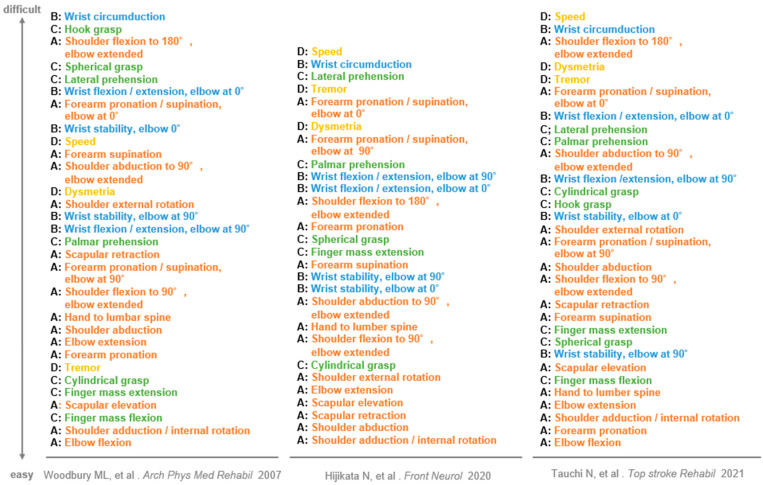
Item difficulty of Fugl-Meyer assessment for upper extremity. The results of three studies examining the difficulty of FMA-UE sub-items are presented. The sub-items with higher difficulty are located toward the top. A, B, C, and D next to the lower-level items indicate the corresponding subsection. As for the participants’ characteristics, Woodbury ML et al. reported that the participants were mainly patients with acute mild hemiplegia. Hijikata et al. reported that the participants were mainly patients with moderate-to-severe chronic-stage hemiplegia. Tauchi et al. reported that the participants were mainly patients with mild-to-moderate subacute-stage hemiplegia. This figure has been partly modified from the articles published by Woodbury, M.L., et al., 2007 [27], Hijikata, et al., 2020 [28], and Tauchi, et al., 2021 [29].

**Figure 4 jcm-11-06835-f004:**
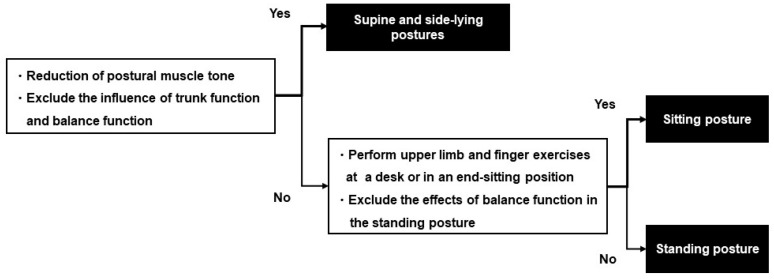
Chart practice posture determination methods. A method for determining the posture of functional exercises is presented. The supine and side-lying postures are chosen when the influence of postural muscle tone reduction, trunk function, and balance function are excluded. The sitting posture is selected when the patient wants to perform upper limb and finger exercises at a desk or in an end-sitting position or when the patient wants to exclude the effects of balance function in the standing posture. The standing posture is selected when the patient’s goal is to perform ADL in the standing posture.

**Figure 5 jcm-11-06835-f005:**
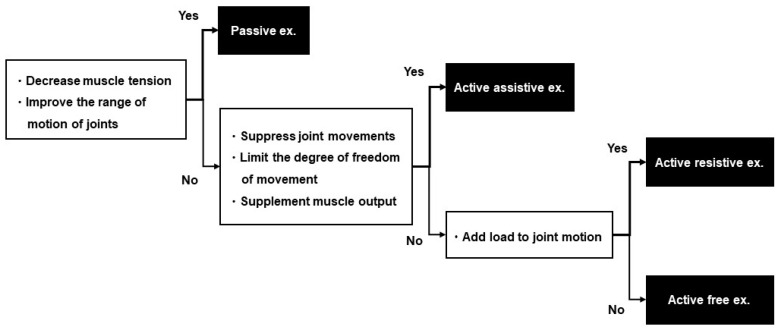
Chart of methods to determine the amount of practice assistance. A method for determining the amount of assistance for performing functional exercises is presented. Passive exercise is selected when the goal is to decrease muscle tension and improve the range of motion of joints. Active assistive exercise is selected when the patient wants to suppress joint movements, limit the degree of freedom of movement, and supplement muscle output. Active resistive exercise is selected to add load to joint motion. Active free exercise is selected to perform upper-extremity movements without adding load to the joint motion.

**Figure 6 jcm-11-06835-f006:**
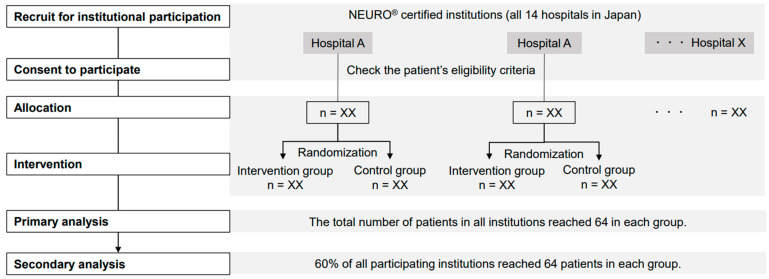
Research procedure. The study setting includes 14 NEURO^®^-accredited facilities in Japan that have agreed to participate in the study. Once patients who meet the study eligibility criteria are enrolled, they will be assigned to the intervention and control groups using a random number table generated for each facility.

**Figure 7 jcm-11-06835-f007:**
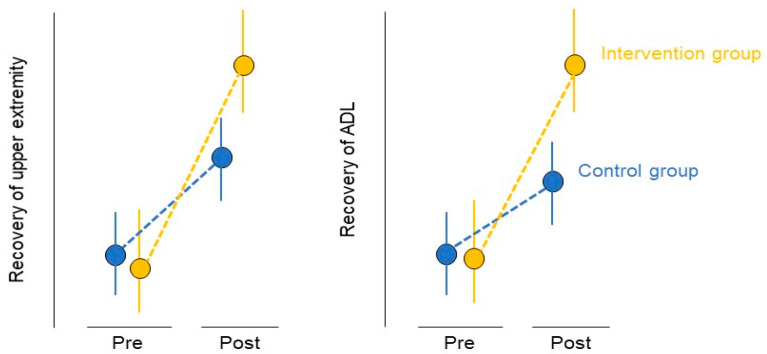
Expected results. ADL, activities of daily living. In the administered occupational therapy, the intervention group is defined by (1) setting the difficulty level of functional exercises, (2) providing an adequate amount of exercise, and (3) the method of goal setting. Therefore, the use-dependent plasticity can be promoted more effectively in the interventional group than in the conventional occupational therapy group, and the patients’ motor functions are expected to improve. Improvement of motor function is expected to contribute to the improvement of patients’ ADL.

**Table 1 jcm-11-06835-t001:** Demographic data of patients.

		Intervention Group	Control Group	Total
Number of patients		N = XX	N = XX	N = XX
Age		XX	XX	XX
Sex		Female = XX Male = XX	Female = XX Male = XX	Female = XX Male = XX
Height		XX	XX	XX
Weight		XX	XX	XX
BMI		XX	XX	XX
Affected side		Left = XX, Right = XX	Left = XX, Right = XX	Left = XX, Right = XX
Dominant hand		Left = XX, Right = XX	Left = XX, Right = XX	Left = XX, Right = XX
Diagnosis	CI	N = XX	N = XX	N = XX
	ICH	N = XX	N = XX	N = XX
Time from onset		XX	XX	XX
FMA-UE severity				
No (<23)		N = XX	N = XX	N = XX
Poor (≤23–≤31)		N = XX	N = XX	N = XX
Limited (≤32–≤47)		N = XX	N = XX	N = XX
Notable (≤48–≤52)		N = XX	N = XX	N = XX
Full (≤53–≤66)		N = XX	N = XX	N = XX

BMI, body mass index; CI, cerebral infarction; ICH, intracranial hemorrhage; FMA-UE, Fugl-Meyer assessment of the upper extremity.

**Table 2 jcm-11-06835-t002:** NEURO^®^ study protocol and momentum settings.

First Week	Monday	Tuesday	Wednesday	Thursday	Friday	Saturday	Sunday
**Event**	Admission						
**rTMS (pulse)**	2400	2400	2400	2400	2400	2400	-
**One-to-one training (time)**	-	500	500	500	500	500	-
**Second week**	Monday	Tuesday	Wednesday	Thursday	Friday	Saturday	Sunday
**Event**						Discharge	
**rTMS (pulse)**	2400	2400	2400	2400	2400	2400	
**One-to-one training (time)**	500	500	500	500	500	-	

rTMS, repetitive transcranial magnetic stimulation. The protocol for admission on Monday is shown. In the columns of one-to-one training, the momentum of joint exercises for the functional exercises set up in this study is shown. Patients will perform the functional exercises of one-to-one training for a total of 1000 times. Patients are exposed to a total of 28,800 rTMS stimuli and 2400 pulses per day, except on Sundays. Patients will be evaluated on the day of admission and on the day of discharge.

**Table 3 jcm-11-06835-t003:** Training menu for each plan.

	One-to-One Training	Self-Training
**Plan 1**	Scapular retraction/protraction Shoulder flexion to 0–180°, elbow extended Shoulder flexion to 0–90°, elbow extended Elbow extension _______________________	Scapular retraction/protraction Shoulder flexion to 0–180°, elbow extended Shoulder flexion to 0–90°, elbow extended Elbow extension ___________________
**Plan 2**	Shoulder flexion to 0–180°, elbow extended Shoulder flexion to 0–90°, elbow extended Elbow extension Finger extension, elbow extended _____________________	Scapular retraction/protraction Shoulder flexion to 0–180°, elbow extendedElbow extension Finger extension ______________________
**Plan 3**	Shoulder flexion to 90–180°, elbow extended Shoulder abduction to 0–180°, elbow extended Wrist flexion/extension, elbow extended Finger extension ____________________	Shoulder flexion to 90–180°, elbow extended Forearm supination/pronation Wrist flexion/extension Finger extension ______________________
**Plan 4**	Shoulder abduction to 0–180°, elbow extended Forearm supination/pronation, elbow extended Wrist flexion/extension, elbow extended Finger extension, elbow extended ______________________	Shoulder flexion to 90–180°, elbow extended Forearm supination/pronation Wrist flexion/extension Finger extension ___________________

Ten exercises are set for each of the four plans, five for one-to-one training and five for self-training. The five exercises in the blank columns are determined by the patient and the therapist based on the patient’s consent, the discussion between the patient and the therapist regarding the exercises that are expected to be highly necessary for the patient to acquire the activities of daily living, and the exercises that meet the patient’s needs. Different practice menus may be introduced for one-to-one training and self-training, but the determined practice plan and the practice menus introduced in the plan should not be changed during the hospitalization period or in the middle of the session. The posture and the amount of load during the exercises are determined according to Figure 4 and Figure 5, respectively, and the occupational therapist may change them according to the patient’s condition and the purpose of the exercises.

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
