# Peer review of "Study Protocol for a Multicenter, Randomized Controlled Trial to Improve Upper Extremity Hemiparesis in Chronic Stroke Patients by One-to-One Training (NEURO®) with Repetitive Transcranial Magnetic Stimulation"

_jcm, 2022, doi:10.3390/jcm11226835_

Round 1

Reviewer 1 Report

Dear Authors,

I have had the pleasure of reviewing your article: „Study protocol for a multicenter, randomized controlled trial to 2 improve upper extremity hemiparesis in stroke patients by occupational therapy with repetitive transcranial magnetic stimulation”.

The article is well written. Very good topic. Results encouraging further work. The results and topic of the work are interesting. The work is worth reading, therefore I recommend it for publication.

Author Response

Response to reviewers’ comments and feedback

Reviewer #1

I have had the pleasure of reviewing your article: „Study protocol for a multicenter, randomized controlled trial to 2 improve upper extremity hemiparesis in stroke patients by occupational therapy with repetitive transcranial magnetic stimulation”.

The article is well written. Very good topic. Results encouraging further work. The results and topic of the work are interesting. The work is worth reading; therefore, I recommend it for publication.

Response to Reviewer #1

Thank you for reviewing our manuscript and for your encouraging comment. We plan to conduct a demonstration experiment in accordance with this protocol and publish the data obtained.

Reviewer 2 Report

It is an interesting protocol study but it need improvement and clarify some points.

In the title, in my opinion, is necessary to add "chronic stroke patients" and remove "occupational therapy" because this therapy is possible do by Physical therapists. I think the title would be "Study protocol for a multicenter, randomized controlled trial to improve upper extremity hemiparesis in chronic stroke patients by one-to-one training (NEURO®)  with repetitive transcranial magnetic stimulation".

The reference 2 is very old, there are news in the bibliography.

Line 37 I think is not very adequate begin a paragraph with "rTMS"

Line 45 what is "convalescent phases"

Study design.  How many arms do you have the study?  and it is an assessor blinded study.

Inclusion criteria. Is this the first stroke? 1) hemiplegia after stroke What scale you performed? What scale dementia is assessed?

3.3. Participants and settings. How many hospitals participate in the study?

3.10. Interventions 

3.10.1. rTMS. and  3.10.2. Rehabilitation these paragraphs are confuse. I think would better the authors explains the intervention and control group in detail.

There is no mention of who is going to train the therapists who are going to carry out the intervention and the evaluators, so that they all carry out the same intervention and the same assessment, that they do it correctly.

Will all patients be irradiated by the same physician? It is not clear.

Line 370. Is the same physician who assessed all patients? it is not clear. “Patients with fever (≥37°C), up- per respiratory tract inflammation, malaise, and taste or olfactory symptoms at the time of examination will be prohibited from admission.” This is an exclusion criteria

Line 375. it is repetitive.

In limitations, the hours and quality of sleep can also be added, since it has been observed that it is important to improve neuroplasticity.

Author Response

Response to reviewers’ comments and feedback

Reviewer #2

It is an interesting protocol study, but it need improvement and clarify some points.

Comment 1

In the title, in my opinion, is necessary to add "chronic stroke patients" and remove "occupational therapy" because this therapy is possible do by Physical therapists. I think the title would be "Study protocol for a multicenter, randomized controlled trial to improve upper extremity hemiparesis in chronic stroke patients by one-to-one training (NEURO®) with repetitive transcranial magnetic stimulation".

Response to comment 1

Thank you for your kind and valuable comment, which has accurately highlighted the shortcomings of our manuscript and helped us improve them. Accordingly, we have revised the title of our manuscript as follows:

“Study protocol for a multicenter, randomized controlled trial to improve upper extremity hemiparesis in chronic stroke patients by one-to-one training (NEURO@) with repetitive transcranial magnetic stimulation”

Comment 2

The reference 2 is very old, there are news in the bibliography.

Response to comment 2

Thank you for pointing it out. As suggested, Reference 2 has been replaced with a recently published paper as shown below:

“2. Palstam, A.; Sjödin, A.; Sunnerhagen, K.S. Participation and autonomy five years after stroke: a longitudinal observational study. PLoS One. 2019, 14, e0219513. doi:10.1371/journal.pone.0219513.” (Lines 477–479)

Comment 3

Line 37 I think is not very adequate begin a paragraph with "rTMS"

Response to comment 3

Thank you for your suggestion. Accordingly, we have revised the sentence as follows:

“The role of rTMS in NEURO® is to pre-condition for rehabilitation by regulating movement-related neural activation.” (Lines 39–40)

Comment 4

Line 45 what is "convalescent phases"

Response to comment 4

Thank you for your question. Convalescent phase refers to the phase with stable symptoms 1 or 2 months after onset. We have added this description in the revised manuscript (lines 47–48).

Comment 5

Study design.  How many arms do you have the study?  and it is an assessor blinded study.

Response to comment 5

Thank you for your question. Since the subjects are hemiplegics, the number of subjects and the number of upper limbs of the paralyzed side are the same. This explanation has been added in the Sample size section (lines 137–138).

Furthermore, the evaluator is not blinded. This explanation has been added in the Blinding section as well (line 158).

Comment 6

Inclusion criteria. Is this the first stroke? 1) hemiplegia after stroke What scale you performed? What scale dementia is assessed?

Response to comment 6

Thank you for your valuable question. We will be selecting patients who had a stroke for the first time, and this has been added to the inclusion criteria. Dementia will be assessed based on scores of the Mini Mental State Examination, which has been included in the manuscript as well (lines 118–120).

Although stroke onset is diagnosed by an emergency physician, we did not specify the scale used in this study.

Comment 7

3.3. Participants and settings. How many hospitals participate in the study?

Response to comment 7

Thank you for your question. Fourteen facilities will participate in this study (line 114).

Comment 8

3.10.1. rTMS. and 3.10.2. Rehabilitation these paragraphs are confuse. I think would better the authors explains the intervention and control group in detail.

Response to comment 8

Thank you for pointing it out. Accordingly, we have included a statement that the methods of rTMS irradiation and occupational therapy session assignment are not changed in both the intervention and control groups (lines 185–186).

Comment 10

There is no mention of who is going to train the therapists who are going to carry out the intervention and the evaluators, so that they all carry out the same intervention and the same assessment, that they do it correctly.

Response to comment 10

Thank you for pointing it out. We have addressed this in the Treatment for the intervention group section as follows:

“The treatment and evaluation procedures for the intervention group will be fully explained to the occupational therapists by at least one expert selected at each site who has discussed this study protocol with the occupational therapists. The expert at each site will supervise the occupational therapists to ensure that they fully understand and implement the study protocol for the intervention group.” (Lines 300–307)

Comment 11

Will all patients be irradiated by the same physician? It is not clear.

Response to comment 11

Thank you for your question. We have included the following sentence in the revised manuscript:

“The rehabilitation physicians affiliated with NEURO®-accredited facilities who are in charge of the patient will administer rTMS.” (Lines 177–178)

Comment 12

Line 370. Is the same physician who assessed all patients? it is not clear. “Patients with fever (≥37°C), up- per respiratory tract inflammation, malaise, and taste or olfactory symptoms at the time of examination will be prohibited from admission.” This is an exclusion criteria.

Response to comment 12

Thank you for your question. We added a description regarding the examining physician to the revised manuscript. In addition, we have included the above statement in quotes as one of the exclusion criteria (lines 122–125).

Comment 13

Line 375. it is repetitive. 

Response to comment 13

Thank you for pointing it out. Line 375 is the bottom line of the Figure legend, and we, therefore, could not identify the sentence you pointed out. However, lines 380–382 were deleted due to repetition (lines 401–403).

Comment 14

In limitations, the hours and quality of sleep can also be added, since it has been observed that it is important to improve neuroplasticity.

Response to comment 14

Thank you for your suggestion. As a secondary assessment, we have added a questionnaire to assess patients’ sleep duration and quality (lines 367–369). In addition, we have added the following report to our bibliography suggesting that sleep affects neuroplasticity (lines 606–608).

“45. Pickersgill, J.W.; Turco, C.V.; Ramdeo, K.; Rehsi, R.S.; Foglia, S.D.; Nelson, A.J. The combined influences of exercise, diet and sleep on neuroplasticity. Front Psychol. 2022, 13, 831819. DOI:10.3389/fpsyg.2022.831819.”

Reviewer 3 Report

Major recommendations:

1. Please consider revising the title. I believe the authors were trying to study standardized OT vs conventional OT in patients undergoing the NEURO rehab program, which in itself has a component of rTMS treatment. The current title is not very clear if the study is to investigate the effect of standardized vs conventional OT or the effect of OT with or without rTMS.

2. Page 3 Lines 112-113: Please specify to what degree will "bilateral cerebrovascular diseases" are excluded. Can patients with bilateral subcortical white matter ischemic changes still participate? How about patients with imaging evidence of prior bilateral lacunar infarcts but never had any clinical presentations?

3. Page 3 Table 1:

- Please consider specifying under CI the locations and etiologies (such as the TOAST criteria [https://www.ahajournals.org/doi/pdf/10.1161/01.STR.24.1.35]) of strokes.

- Under ICH, please also specify the types and locations of hemorrhages.

- How about hemorrhagic transformation after ischemic infarcts? How is that going to be classified? (Consider the Heidelberg Bleeding Classification)

- Also, please collect the NIHSSs [https://www.stroke.nih.gov/resources/scale.htm] and ICH scores [https://www.mdcalc.com/calc/402/intracerebral-hemorrhage-ich-score] for participants and consider using them as additional outcome measurements.

4. Page 4 Lines 145-150: Please test the success of blinding among the patients.

5. Page 5 Lines 87-194: Please explain if the control group will have the same total amount of time of rehab exercises (or total number of functional exercises) compared to the intervention group.

6. Page 11 Line 376: Has this study already begun?

Minor recommendations:

1. Page 5 Line 162: Please rephrase "irradiated".

Author Response

Response to reviewers’ comments and feedback

Reviewer #3

Comment 1

Please consider revising the title. I believe the authors were trying to study standardized OT vs conventional OT in patients undergoing the NEURO rehab program, which in itself has a component of rTMS treatment. The current title is not very clear if the study is to investigate the effect of standardized vs conventional OT or the effect of OT with or without rTMS.

Response to comment 1

Thank you for pointing it out. We have revised the title accordingly.

Comment 2

Page 3 Lines 112-113: Please specify to what degree will "bilateral cerebrovascular diseases" are excluded. Can patients with bilateral subcortical white matter ischemic changes still participate? How about patients with imaging evidence of prior bilateral lacunar infarcts but never had any clinical presentations?

Response to comment 2

Thank you for your questions. According to the Japanese rTMS guidelines, bilateral cerebrovascular disease is excluded from the treatment. Therefore, we excluded these patients from the study regardless of their symptoms due to bilateral cerebrovascular disease (Participants and settings section, lines 116–118).

Comment 3

  1. Page 3 Table 1:Please consider specifying under CI the locations and etiologies (such as the TOAST criteria [https://www.ahajournals.org/doi/pdf/10.1161/01.STR.24.1.35]) of strokes.

Under ICH, please also specify the types and locations of hemorrhages.

How about hemorrhagic transformation after ischemic infarcts? How is that going to be classified? (Consider the Heidelberg Bleeding Classification)

Also, please collect the NIHSSs [https://www.stroke.nih.gov/resources/scale.htm] and ICH scores [https://www.mdcalc.com/calc/402/intracerebral-hemorrhage-ich-score] for participants and consider using them as additional outcome measurements.

Response to comment 3

Thank you for your questions and suggestions.

We plan to obtain information on the foci of cerebral infarction and cerebral hemorrhage. Regarding the classification of the disease type, there is a classification system proposed by Reviewer #3; however, previous studies have shown that there is no difference in NEURO@ outcomes between patients with cerebral infarction and cerebral hemorrhage (Tatsuno et al., 2021). We have chosen not to specify the type of stroke as a confounding factor in our protocol.

Reference #24

Tatsuno, H.; Hamaguchi, T.; Sasanuma, J.; Kakita, K.; Okamoto, T.; Shimizu, M.; Nakaya, N.; Abo, M. Does a combination treatment of repetitive transcranial magnetic stimulation and occupational therapy improve upper limb muscle paralysis equally in patients with chronic stroke caused by cerebral hemorrhage and infarction?: a retrospective cohort study. Medicine (Baltimore). 2021, 100, e26339. DOI:10.1097/MD.0000000000026339.

As Reviewer #3 pointed out, the NIHSS is a valid assessment outcome. It has two items on upper extremity motor function, which we speculated would have a ceiling effect if used as the outcome for this study. Although we did carefully consider this, we did not use the NIHSS in this study and instead planned to use the Fugl-Meyer assessment of the upper extremity, the Action Research Arm Test, and the Wolf Motor Function Test, which can assess motor function in detail. Function Test can assess detailed motor function.

Comment 4

  1. Page 4 Lines 145-150: Please test the success of blinding among the patients.

Response to comment 4

Thank you for your suggestion. In this study design, patients will be blinded to the treatment they receive, as described in the following sentence:

“In order to confirm the success or failure of the patients’ blinding, therapists will ask patients which treatment they thought they received after completing NEURO treatment and then confirm whether it was the treatment actually performed.” (Lines 159–162)

Comment 5

  1. Page 5 Lines 87-194: Please explain if the control group will have the same total amount of time of rehab exercises (or total number of functional exercises) compared to the intervention group.

Response to comment 5

Thank you for your question. The practice time for both the intervention and control groups is the same. Also, in line with Reviewer #2’s opinion, a supporting reference has been added to the manuscript (lines 197–200). However, we did not set the total number of functional exercises for the control group; therefore, we cannot compare the total number of functional exercises between both groups. Although we considered logging the patients’ arms with a vibrometer to measure the number of exercises, we did not go further with the idea in this study due to a lack of research funds. Nonetheless, this is an important comment that will have to be resolved eventually.

Comment 6

  1. Page 11 Line 376: Has this study already begun?

Response to comment 6

Thank you for your question. The study has not yet started and will begin after this protocol manuscript is accepted. We have revised the text as follows:

“This study will begin in December 2022 and end in August 2024.” (Lines 399)

Comment 7

Page 5 Line 162: Please rephrase "irradiated".

Response to comment 7

Thank you for your suggestion. Accordingly, we have revised the sentence as follows:

“The rTMS will be directed at the primary motor cortex of the patients’ healthy cerebral hemisphere at 2,400 pulses a day at a low frequency of 1 Hz, as in the conventional NEURO® protocol (Table 2).” (Lines 174–177)

We appreciate all the comments and suggestions provided to improve our manuscript and sincerely thank all the reviewers.
